# Harnessing Query Heterogeneity for Cost-Effective Proactive Caching in LLM Inference

## Abstract

As Large Language Models (LLMs) significantly enhance the capabilities of AI systems, the increasing volume of query processing requests presents challenges for cost-effective inference, particularly due to repetitive queries that lead to unnecessary resource consumption and increased costs. Caching strategies are employed to store a small set of previous queries, enabling direct retrieval of repetitive queries without reprocessing by the LLMs. However, existing caching algorithms often assume uniform query lengths, simplifying cache selection to a top-$K$ problem, which is inadequate for real-world scenarios with heterogeneous lengths. To address this issue, we propose a bandit learning algorithm for proactive query caching in LLMs, specifically considering variable-sized queries. We cast the optimal cache query cache problem as a knapsack problem. Since the repetitive pattern and processing cost are unknown and has uncertainty, we cast the learning-to-cache problem as a bandit learning problem. Compared to conventional bandit learning frameworks, a new technical challenge is that the reward of an arm would not be observed if it is pulled. To tackle this, we propose an Lower confidence bound (LCB)-type algorithm, which we prove has a $\tilde{O}(\sqrt{T})$ order of regret and show that our regret does not deteriorate compared to previous results when incorporating a variable size setting. Furthermore, we demonstrate that our online cache policy effectively reduces the additional computational overhead typically associated with calculating the optimal cache.

## 1 Introduction

Large Language Models (LLMs) have gained increasing popularity across various fields (Zhang et al., 2024; Wu et al., 2023b;a). However, as the capabilities of LLMs have increased, the challenge of resource consumption during LLM deployment has become increasingly significant and cannot be ignored. This leads to higher inference costs and longer latency compared to traditional models. As a result, when a large number of queries must be processed simultaneously, LLMs often prove to be inefficient or unsuitable for such tasks (Chen et al., 2023).

In the real-world marketplace of LLMs, it is inevitable for online large model systems to handle some repetitive tasks (Chen et al., 2023). For the same task, the large model needs to be called repeatedly multiple times to obtain the same result, which will inevitably lead to a waste of resources. As the scale of LLMs grows larger, the cost of repeated queries becomes increasingly significant. Developing methods to reduce these redundant costs and resource waste would be highly beneficial for the efficient deployment of large models, significantly lowering the overall operational costs associated with them.

In the field of computer memory access (Belady, 1966), caching is an effective method for optimizing data access speed and reducing the overhead of resource redundancy by establishing a fast retrieval mechanism within the system. When applied to large model systems, caching allows repeated queries to directly retrieve results from the cache without requiring complex computations from the LLMs, as the results of these repeated queries are assumed to remain largely unchanged.

Recent research (Stogiannidis et al., 2023; Zhu et al., 2024) involve a cache to store and reuses LLM API's response when similar queries are repeatedly asked. These works have successfully integrated caching strategies into the deployment of large models, but to cope with more complex real-world

online environments, there is still much room for improvement in the field of cache combined with LLMs. Incorporating cache into LLMs presents several challenges: 1) how to determine which query to cache appropriately, 2) managing incomplete information when updating the cache in large-scale, dynamic user environments. For the first challenge, in order to reduce overhead, we need to cache those queries that may be popular and incur high costs. However, the cost associated with each query may be unknown and variable; it may vary depending on different users or changes in the computation process. From a statistical perspective, the cost of processing each query is a random variable that depends on the query itself and fluctuates due to variations in the auto-regressive generation process, as well as the length and quality of the generated responses (Zhu et al., 2024). For the second challenge, since the cost of each query may depend on the preferences of different users or other unpredictable factors, and the popularity of queries shifts over time, it is impossible to know all relevant information for each query in advance. This requires a strategy that dynamically collects specific statistics for each query through feedback in an online environment to update the cache.

To deal with the selection of queries, Zhu et al. (2024) takes into account both the expected cost and popularity of queries as criteria for cache selection. This natural approach has proven effective in experiments and is relatively easy to implement. However, Zhu et al. (2024)'s online algorithm performs well only when each query has the same and fixed size in the cache, meaning the number of queries that can be stored in the cache remains constant. In the case where different queries have variable size, although they make a simple discussion in appendix, there still exists room for improvement. In variable size case, in addition to considering the cost and popularity of a query, the length of the query must also be taken into account. At this point, the number of queries that the cache can store will no longer be constant.

In this paper, we consider the aforementioned issue from the perspective of bandit with knapsack constraint. We propose an online algorithm **Online Stream Cache Bandits**, an extension variant of Zhu et al. (2024) to solve the problem. Our main contributions are as follows:

**Algorithmic Framework.** The proposed Online Stream Cache Bandits algorithm contains four key components. 1) *Selecting cache with knapsack constraints*: We consider the issue of variable size as a 0-1 knapsack problem. Here, the cost and popularity of a query represent the value, while the size of the query corresponds to its weight. Thus, selecting an optimal cache is equivalent to solving for the optimal knapsack configuration. 2) *Lower confidence bound bandits*: Different from traditional bandit literature using UCB-type algorithms, we use lower confidence bound for estimation. This choice is driven by the fact that, in our environment, the selected arm is not always observed in each round due to cahce hit, which incurs zero cost (Zhu et al., 2024). On the other hand, this lower confidence bound acts like a pessimistic strategy, helping to avoid our cache policy trapping into sub-optimal solution. 3) *Double scheduling strategy*: The introduction of the knapsack problem into our policy incurs additional computational overhead. If the knapsack problem is trivially computed in *each* round, this additional computational overhead will grow linearly. To mitigate this, we use a double scheduling strategy, where we only invoke the knapsack problem solver to update the whole cache at rounds that are integer powers of 2, reducing the additional cost to logarithmic order. 4) *Recommended cache*: Using the trivial method mentioned above, extra memory space is required to store all historical information and result of each query so that it can be selected in a timely manner when updating the cache. However, this extra memory space is often impractical in real-world scenarios. To address this, we introduce a recommended cache that temporarily stores the optimal cache selecting identified by the knapsack problem and "waits" for the relevant query. This approach eliminates the need for additional memory space, and we show that this modification does not affect the regret bound.

**Theoretical Analysis.** The involving of the knapsack problem presents several challenges in our theoretical analysis, such as we can not determine how many queries are currently stored in the cache or to find the regular of value of the queries stored in the current cache. Despite these challenges, we demonstrate that our Online Stream Cache Bandits algorithm achieves a $\tilde{O}(\sqrt{T})$ regret bound with a $O(\log T)$ additional computational cost. This result aligns with the findings of Zhu et al. (2024)'s work, which is the most relevant work to ours, and the order of $T$ in our regret bound aligns with their established lower bound of $\tilde{O}(\sqrt{T})$. Furthermore, we show that the work of Zhu et al. (2024) can be seen as an special case of ours when all queries are set to the same length.

**Experiments Evaluation.** Zhu et al. (2024) proposed a heuristic method that replaces one query with the smallest expected cost per-size, defined as $P(q)C^*(q)/S(q)$, each round to solve variable size case. However, their method requires a restrictive condition: the length of the query with the result must be much smaller than the length of the cache, *i.e.* $S(q) << M$. Our analysis shows that their method will fail without this condition. Using Zhu et al. (2024) as our baseline, we conduct experiments on both synthetic and real data. Our results show that the Online Stream Cache Bandits algorithm consistently achieves superior performance in terms of regret across both simulation and real-world datasets.

## 2 RELATED WORK

In the field of LLM, cache technology can address the issue of repeated inquiries, thereby reducing the cost of the LLM marketplace (Chen et al., 2023). Recently, caching strategies tailored for LLMs have emerged (Xu et al., 2023; Zhang et al., 2023), focusing on content reuse during the inference phase. These strategies aimed to optimize the internal data flow within LLMs, minimizing delays and lowering resource consumption. Bang (2023) is recognized in the industry for its simplicity and flexibility, providing various options for integration with different LLMChat services. However, it does not offer advanced cache performance enhancements. In contrast, Li et al. (2024) focuses on semantic-oriented enhancement, they improve cache efficiency by leveraging the semantic understanding of queries. Zhu et al. (2024) propose a Caching in Online Learning algorithm to avoid resource wasting. In their further discussion, they mention that generalizing their work to a variable-size cache, in which distinct queries have heterogeneous lengths. However, they did not conduct detailed research or provide a rigorous analysis. Inspired by Zhu et al. (2024), we generalize their problem with a variable size constraint. Furthermore, we consider involving the knapsack problem to solve this variable size issue and propose the algorithms with theory guarantees.

Cache is a key technology that enhances system performance by reducing the overhead of redundant computations. Traditional cache replacement algorithms delve into the most effective strategies for caching queries that vary in frequency, cost, and required cache space. To address the issue of varying frequencies, Lee et al. (2001) propsed a standard approach which use a Least Frequently Used (LFU) or Least Recently Used (LRU) cache eviction strategy, and these work have been proven to be optimal for both adversarial and stochastic queries (Bura et al., 2022). When varying costs and varying frequencies exist simultaneously, Jin & Bestavros (2000), Arlitt et al. (2000) propose and study the Greedy Dual-Size with Frequency (GDSF) replacement algorithm, which takes both frequency and cost into consideration. Our problem setting may provide inspiration for solving cache problems that involve uncertainty.

The Knapsack Problem (Kellerer et al., 2004) is a classic problem in computer science and optimization theory. In the algorithms we propose, the knapsack problem will determine how to select the cache efficiently. We can regard our problem as a 0-1 knapsack problem and use dynamic programming (Bertsekas, 2005), a classical method to address the knapsack problem, as our Oracle to update our policy. The online knapsack problem (Zhou et al., 2008) is an online variant of the knapsack problem, which requires making decisions on the spot without knowing all the inputs. The challenge of the online knapsack problem lies in making optimal or near-optimal decisions under incomplete information, which often requires designing effective algorithms to approximate the optimal solution.

The multi-armed bandit problem is a reinforcement learning problem that has broad applications in finance (Shen et al., 2015), medicine (Liu et al., 2018), chemistry (Wang et al., 2024) and recommendation systems (Zhou et al., 2017). In the classic stochastic formulation, a casino player is faced with a set of slot machines, each with a fixed but different reward distribution that is initially unknown. With a limited budget, the player's goal is to maximize the total payout by identifying and playing the slot machines that offer better returns. To this end, the player effectively allocates limited resources to balance exploration of the machines that have been played less and exploitation of the current best options. In our work, we can see each query as an arm, and its cost is an unknown distribution. Under this setting, the UCB-type method can deal with the uncertainty of each query. Compared to Bandits with Knapsacks (BwK), which is a problem that combines bandit and knapsack constraints, our problem setting is different. The BwK problem (Badanidiyuru et al., 2018; Liu et al., 2024) assumes that there are $K$ resources consumed over time, each with distinct

budgets $B_1, ..., B_K$. Every resource $i \in [K]$ is associated with a consumption, The goal of policy is to maximizes the accumulated reward subject to the budget constraints that each consumption of resource $i$ is less than $B_i$. In our work, we utilize the knapsack problem to address heterogeneous size queries, treating the cache as a knapsack constraint: the lengths of distinct queries represent the weights, while the costs and frequencies represent the values. We impose a fixed budget constraint in each round, this setting is similar to Yu et al. (2016). Compare to their work, our approach incorporates missing feedback, which involves greater challenges and more complex analysis.

## 3 SIZE-AWARE CACHING BANDIT SETTING

In this section, we formulate the size-aware caching bandit problem in LLM inference serving with heterogeneous queries.

**Query and request cost.** We consider a finite set of queries $\mathcal{Q} = \{q_1, ..., q_N\}$ containing $N$ distinct queries, each query $q \in \mathcal{Q}$ is associated with a size $L(q) \in \mathbb{R}_+$. When query $q$ is input to the LLM system, the LLM processes it and returns a corresponding result $r(q)$ with size $A(q) \in \mathbb{R}_+$. We use $S(q) = L(q) + A(q)$ to denote the *total-query size* of $q$. Every time the LLM processes $q$, it will incur a random cost $C(q)$ with unknown mean $C^*(q)$:

$$C(q) = C^*(q) + \epsilon_q$$

where $\epsilon_q$ is a sub-Gaussian noise that captures the uncertainties in the cost, with $\mathbb{E}[\epsilon_q] = 0$. Remark that this setup is reasonable as most noise in real-world applications follows a sub-Gaussian distribution. Consistent with Zhu et al. (2024), the cost in a LLM can be FLOPS, latency of the model, the price for API calls, user satisfaction of the results, or a combination of all these factors.

**Size-aware cache.** To save the cost of repeatedly processing the queries, we maintain a cache $\mathcal{M}$ for the LLM system, storing a small subset of queries with their corresponding results. The size of cache has a maximum size $M$, the total size of the stored queries must satisfy $\sum_{q \in \mathcal{M}} S(q) \leq M$. Let $\mathfrak{J}$ be the set of all possible caches that satisfy the size constraint

$$\mathfrak{J} = \{\mathcal{M} | \sum_{q \in \mathcal{M}} S(q) \leq M\}$$

Because the cache maintains the query and its result, when a query hits the cache, it will output the result of matched query directly with zero cost. Previous works such as Zhu et al. (2024) assume that the size of queries is homogeneous means that $S(q) = 1$ for all queries, reducing the the problem to a top-$K$ selection problem. We instead consider a heterogeneous setting, where queries have different sizes $S(q)$, introducing additional challenges. This heterogeneity is a key difference between our model and previous works such as Zhu et al. (2024).

**Online learning size-aware caching bandit.** We consider a total number of $T \in \mathbb{N}_+$ learning rounds. In each round, a query arrives, which is sampled from $\mathcal{Q}$ according to a fixed unknown population distribution $\mathbb{P} \in \Delta(\mathcal{Q})$. Let $q_t \in \mathcal{Q}$ represent the query sampled from $\mathcal{Q}$ in round $t$. We use $P(q) \in (0, 1]$ to denote the probability that the query $q$ be selected in each round, such that $\sum_{q \in \mathcal{Q}} P(q) = 1$.

In round $t$, the agent first selects *current cache* $\mathcal{M}_t$, which satisfies the size constraint, based on the history information from previous rounds. After $q_t$ is sampled from $\mathbb{P}$, the agent will first check the current cache $\mathcal{M}_t$ and If the query $q_t$ is found in the cache, i.e., $q_t \in \mathcal{M}_t$, we say the query *hits* the cache. In this case, the result of $q_t$ is directly returned without further processing by the LLM. The cost of processing this query is $0$ and will save a potential cost $C(q_t)$, which is unobserved to the agent. If query $q_t$ does not hit the cache, the system processes the query, incurring a cost $C(q_t)$, and returns the result.

The objective is to choose the optimal cache without knowing the cost and popularity of each query. To achieve this goal, one can use a bandit-type policy to deal with the uncertainty, treating each query as an arm in a multi arm bandit model. However, our setting differs from traditional bandit problems because feedback (i.e., the cost) is not observed in every round. Specifically, when a query hits the cache, no cost feedback is received. Only when a query misses the cache does the agent observe a random cost as feedback. This "reverse-bandit" scenario, where feedback is only available when an arm is not selected, complicates the application of traditional bandit algorithms.

**Learning objective.** Our goal is to minimize the total cost with time horizon $T$. From another perspective, the cost we saved by cache hitting can be regarded as additional reward we have obtained. Following Zhu et al. (2024), we can define the reward of a query $q$ with a given cache $\mathcal{M}$ as

$$f(\mathcal{M}, q) = C(q)\mathbb{I}\{q \in \mathcal{M}\}$$

And the regret can be defined as

$$\text{Reg}(T) = \mathbb{E}\left[\sum_{t=1}^{T} f(\mathcal{M}^*, q_t) - f(\mathcal{M}_t, q_t)\right] = \mathbb{E}\left[\sum_{t=1}^{T} C(q_t)\left(\mathbb{I}\{q_t \in \mathcal{M}^*\} - \mathbb{I}\{q_t \in \mathcal{M}_t\}\right)\right] \quad (1)$$

where $\mathbb{I}$ is the indicator function and $\mathcal{M}^*$ is the optimal cache.

To minimize the total cost, we need to find the optimal cache $\mathcal{M}^*$ which maximizes the saved cost. By taking expectation over all query, the expected reward function can be defined as:

$$F(\mathcal{M}, \boldsymbol{C}^*, \boldsymbol{P}) = \mathbb{E}\left[f(\mathcal{M}, q)\right] \quad (2)$$

$$= \mathbb{E}\left[C(q)\mathbb{I}\{q \in \mathcal{M}\}\right] \quad (3)$$

$$= \sum_{q \in \mathcal{M}} C^*(q)P(q) \quad (4)$$

where $\boldsymbol{C}^* = \{C^*(q)\}_{q \in \mathcal{Q}}$ is the set of unknown costs for all queries, and $\boldsymbol{P} = \{P(q)\}_{q \in \mathcal{Q}}$ represents the probability of each query being sampled.

We see this expectation of reward function as the reward of a cache $\mathcal{M}$. The optimal cache can therefore be formally written as $\mathcal{M}^* = argmax_{\mathcal{M} \in \mathfrak{I}} F(\mathcal{M}, \boldsymbol{C}^*, \boldsymbol{P})$ and the regret of the policy can be expressed as:

$$\text{Reg}(T) = \mathbb{E}\left[\sum_{t=1}^{T} F(\mathcal{M}^*, \boldsymbol{C}^*, \boldsymbol{P}) - F(\mathcal{M}_t, \boldsymbol{C}^*, \boldsymbol{P})\right] \quad (5)$$

We model this as a 0-1 knapsack problem where $C^*(q)P(q)$ is the value of $q$ and $S(q)$ is its weight. The cache is treated as a knapsack with capacity of $M$, and the optimal cache selection corresponds to solving for the optimal knapsack. In order to adapt the knapsack problem to our model, we need the following assumption:

**Assumption 1.** *The cost can be bounded by $c_1 \leq C^*(q) \leq c_2, \forall q \in \mathcal{Q}$, where $c_1, c_2 \in \mathbb{R}^+$ and $c_2 > c_1$*

**Assumption 2.** *The popularity can be bounded by $0 < P(q) < 1/2, \forall q \in \mathcal{Q}$.*

**Offline size-aware cache provisioning.** As mentioned above, finding the optimal cache can be viewed as solving a 0-1 knapsack problem. Therefore, a solver for the knapsack problem will be of significant help in selecting the optimal set of queries to cache. To do this, we introduce an Oracle for solving the knapsack problem, which takes the weight and size of each query as inputs and returns the optimal knapsack configuration. Specifically, We use $Oracle(\mathcal{Q}, \boldsymbol{w}, \boldsymbol{l}, M) \longrightarrow \mathcal{Q}_{\boldsymbol{l}, M}^{\boldsymbol{w}*}$ to yield the optimal knapsack of knapsack problem with volume constraint of $M$, where $\mathcal{Q}$ is the input query set, $\boldsymbol{w} = \{w(q)\}_{q \in \mathcal{Q}}, \boldsymbol{l} = \{L(q)\}_{q \in \mathcal{Q}}$ denote the weight vector and value vector of each query in $\mathcal{Q}$ respectively. The Oracle solver (e.g., Dynamic Programming) returns the optimal knapsack based on the length and the product of estimated costs and selection probabilities for all queries.

**Comparison with previous work.** Unlike Zhu et al. (2024), where the cache size refers to the number of queries stored, in our setting, it refers to the number of tokens the cache can store. Here, $M$ represents the total token capacity of the cache. The values $L(q)$ and $A(q)$ correspond to the token counts for the query $q$ and its result, respectively, which together represent the cache size required for $q \in \mathcal{Q}$. When a query $q$ is input, it is tokenized with a length $L(q)$, representing the number of tokens the query contains. The LLM processes this query and produces a result with length $A(q)$. Therefore, the total size required in the cache for query $q$ is $S(q) = L(q) + A(q)$. In our setting, the cache constraint at each round $t$ must satisfy: $\sum_{q \in \mathcal{M}_t} S(q) \leq M$ in each round $t$.

We also note that in the online model, we discard the results of queries that are not stored in the cache, but we maintain estimation information such as the query's length and cost. If a previously dropped query reappears, the result must be recomputed by the LLM. This setup reflects real-world conditions where the number of queries is large and there is insufficient storage to keep all query results in the cache.

# 4 ALGORITHM

In this section, we propose our batch and stream version algorithm to address the cache selecting with variable size query. Because the algorithm has not seen any queries at the beginning, it needs to record the indices of queries to maintain the estimation for each query in cache selection. We use a set $Q_t$ in our algorithm to represent this record, a query $q \in Q_t$ means that the estimation and index information of this query have been recorded by algorithm.

## 4.1 BASIC BATCH CACHE BANDITS

**Batch setting.** We begin our algorithm with a batch setting, which means that the algorithm can record all the queries along with their results and can obtain any query with its result at any time. In this case, a straightforward policy for our model is that we can naturally use the Oracle at the end of each round, and the agent selects the cache according to the results provided by the Oracle. We first propose a batch version of our policy (Algorithm.1), this version does not consider the storage costs of all queries and their results, and it assumes that the results of the required queries can be obtained at any time. In this algorithm, the cache will be reselected in each round. At the end of each round, the Oracle will provide the optimal solution based on the current estimation of all queries; then the policy will select queries according to the output of the Oracle and add them to the cache. Under Algorithm.1, the current cache will ultimately converge to the optimal cache when the estimations converge to the true values. On the other hand, if the estimations of all queries become precise enough, the current cache will be updated to the optimal cache immediately.

**Lower confidence bound.** Similar to Zhu et al. (2024), we also use a lower confidence bound in the estimation of the average cost of each query; this differs from traditional UCB-based methods. The reason we use the lower bound is that the pessimistic policy in cost will prevent the algorithm from getting trapped in a sub-optimal solution. In our setup, the optimal cache always selects the query set with the largest value sum, which is the sum of $C^*(q)P(q)$ in our model. If we use an upper confidence bound for estimation, once the Oracle outputs a sub-optimal solution for cache selection, it will maintain the estimation of queries included in this sub-optimal cache as greater than that of other queries with high probability. This prevents the algorithm from correcting the solution selected by the Oracle, leading to a trap in the sub-optimal cache. This is akin to adding a penalty to the estimation of queries that remain in the cache for a long time, making them more likely to be replaced, similar to an enforcement of exploration.

**Replacement strategy.** In Zhu et al. (2024), they proposed a method that replacing one query with the smallest expected cost per-size $P(q)C^*(q)/S(q)$ each round. This method is natural and does not incur any additional cost. However, in certain situations related to the knapsack problem, due to the size constraint, the current cache may need to replace multiple queries at once to update to the optimal solution. If we use the LFU policy and allow only one query to be replaced at a time, it may prevent the algorithm from updating to the optimal cache. Unlike Zhu et al. (2024), the Oracle we use here allows the policy to update more than one query each round, which can prevent the algorithm from getting trapped in a sub-optimal solution.

**Practical limitations.** Although Algorithm 1 can effectively select the optimal solution, it has significant limitations. Algorithm 1 requires a record of all queries, and their results must be obtainable at any time to ensure it can use the appropriate queries to update the cache promptly. Thus, we need a large amount of additional storage space to keep all the information about queries and their results; however, this is not feasible in an online setting. In fact, this additional storage requirement is unreasonable in real-world scenarios. On one hand, if we could store all the results of queries, the cache would become unnecessary. On the other hand, Algorithm 1 needs to call the Oracle in each round, which incurs a significant additional computational cost as the number of rounds increases, especially in solving the NP-hard problems like the knapsack problem, such a high computational cost is unsustainable.

## 4.2 ONLINE STREAM CACHE BANDITS

To address the issue of additional storage and computational cost, we propose an online version of our policy (Algorithm 2).

---

**Algorithm 1** Basic Batch Cache Bandits

---

**Require:**

$\mathcal{Q}_0 = \emptyset$. $T_0^{(p)}(q) = 0, q \in \mathcal{Q}$. $T_0^{(c)}(q) = 0, q \in \mathcal{Q}$. $\hat{P}_0(q) = 0, q \in \mathcal{Q}_t$. $\tilde{C}_0(q) = 0, q \in \mathcal{Q}$. $\hat{C}_0(q) = 0, q \in \mathcal{Q}$. $A(q) = 0, q \in \mathcal{Q}$. Current cache $cache_0 = \phi$. The size of cache is $M$.

**Ensure:**

1: **for** round $t = 1, ..., T$ **do**
2:     A user arrive and select $q_t$ sampled from $\mathcal{Q}$ with the length of query is $L(q_t)$
3:     **if** $q_t \notin \mathcal{Q}_{t-1}$ **then**
4:         $\mathcal{Q}_t = \mathcal{Q}_{t-1} \cup \{q_t\}$
5:         Recording the output result of $q_t$ when $q_t$ is input into the LLM in the following.
6:     **else**
7:         $\mathcal{Q}_t = \mathcal{Q}_{t-1}$
8:     **end if**
9:     $T_t^{(p)}(q_t) = T_{t-1}^{(p)}(q_t) + 1$
10:    $T_t^{(p)}(q) = T_{t-1}^{(p)}(q), \; \forall q \neq q_t$
11:    $\hat{P}_t(q) = T_t^{(p)}(q)/t, \; \forall q \in \mathcal{Q}_t$
12:    **if** $q_t \in \mathcal{M}_{t-1}$ **then**
13:        Output the result from cache
14:        $\hat{C}_t(q) = \hat{C}_{t-1}(q), \tilde{C}_t(q) = \tilde{C}_{t-1}(q), T_t^{(c)}(q) = T_{t-1}^{(c)}(q), \forall q \in \mathcal{Q}_t$
15:    **else**
16:        Agent input $q_t$ into LLM and receive the result of query with length $A(q_t)$, then observe the cost $c_t(q_t)$.
17:        $\tilde{C}_t(q_t) = \tilde{C}_{t-1}(q_t) + c_t(q_t), T_t^{(c)}(q_t) = T_{t-1}^{(c)}(q_t) + 1$
18:        $\hat{C}_t(q) = \hat{C}_{t-1}(q), \tilde{C}_t(q) = \tilde{C}_{t-1}(q), T_t^{(c)}(q) = T_{t-1}^{(c)}(q), \forall q \neq q_t$
19:        $\hat{C}_t(q) = max\{c_1, \frac{\tilde{C}_t(q)}{T_t^{(c)}(q)} - (c_2 - c_1)\sqrt{\frac{2\log(6TN/\delta)}{T_t^{(c)}(q)}}\}, \; \forall q \in \mathcal{Q}_t$
20:    **end if**
21:    $\boldsymbol{w}_t = \{\hat{C}_t(q) \cdot \hat{P}_t(q)\}_{q \in \mathcal{Q}_t}, \boldsymbol{l}_t = \{L(q) + A(q)\}_{q \in \mathcal{Q}_t}$
22:    $Oracle(\mathcal{Q}_t, \boldsymbol{w}_t, \boldsymbol{l}_t, M) \to \mathcal{M}_t$
23:    Putting those queries with result which belong to $\mathcal{M}_t$ into cache from record.
24: **end for**

---

**Stream-Based Cache Updates.** In the online model, the algorithm operates in a stream-based manner, meaning it does not store all previous queries and their results. Instead, it maintains only the sequence numbers and estimated values of the queries. Whenever the cache is updated, any queries and their corresponding results that are removed from the cache are permanently cleared. As a result, the algorithm can only access the queries currently in the cache and any newly arriving queries. It may not be able to update the cache immediately based on the output of the oracle because the required queries may have already been cleared due to replacement, necessitating the re-acquisition of their results when they arrive next. This resembles a data stream, where the algorithm can only see the content of newly arrived queries and cannot access the content of all queries in each round. To solve this issue, we involve a recommended cache denoted by $\hat{\mathcal{M}}$ to temporarily store the results output by our Oracle at the current time.

When Algorithm 2 call the Oracle, our policy will not update cache immediately, instead it will first get a recommended cache $\hat{\mathcal{M}}_t$, and clear out the current cache. In the following rounds, if a query that belongs to the recommended cache arrives, it will miss the current cache and be input into the LLM to get its result. Then, this query will be put into the current cache; otherwise, no update is made to the cache, regardless of whether the current query hits. This process will repeat until the next recommended cache update occurs. Under this replacement method, we do not need to store all results of each queries, which satisfy the online model and be more reasonable.

**Reducing computation.** To reduce the additional computational cost, we establish a special protocol which only call the Oracle in the round that is the power of 2, this protocol can reduce the computational cost of calling Oracle from $O(T)$ to $O(\log T)$.

Although the number of cache misses seems to increase due to the lack of timely updates in the online mode and the reduction of Oracle calls to minimize computational overhead, we can show that the deterioration of the overall algorithm is not significant. Unlike traditional bandit algorithms, the update for the estimation of a query cost occurs only when the query does not hit the cache. Therefore, if the number of misses increases in the previous round, the estimation of the query will become more precise afterward, leading the knapsack returned by the oracle to gradually approach the optimal knapsack. In other words, the more misses there are, the more precise the query estimation will be. In the same time, the times of clear operation will be bound by $O(\log T)$, which further bound the number of missing.

Algorithm 2 balances the computational cost and the convergence speed. In the batch version, the current cache updates immediately to the optimal cache when the oracle provides the optimal knapsack. However, it needs to call the oracle in each round to ensure the algorithm knows the optimal solution in a timely manner, which leads to an additional computational cost of $O(T)$. In the online version, we do not need to call the Oracle in each round, and we can demonstrate that the regret is still bounded by $\tilde{O}(\sqrt{T})$.

### 4.3 REGRET ANALYSIS

For the knapsack problem, let $\ell(M, \mathcal{Q}) = \max_{\mathcal{M} \in \mathfrak{J}} |\mathcal{M}|$ represent the maximum number of queries that the optimal knapsack can contain with capacity $M$, where $\mathcal{Q}$ is the query set with associated values and weights. Then, in our setting, $\ell(M, \mathcal{Q})$ gives the maximum number of query that the cache can store. We have the following regret guarantee for our algorithms:

**Theorem 1.** *For the Basic Batch Cache Bandits algorithm ( Algorithm 1), assuming sufficient memory space to store results of all queries,and setting $\delta = 1/T$, the regret is upper bounded by*

$$Reg(T) \leq O\left(\frac{\ell(M, \mathcal{Q})}{p^*} \sqrt{T} \log^{\frac{3}{2}}(TN)\right) \tag{6}$$

*where $p^* = \min_{q \in \mathcal{Q}} P(q)$ is the minimum popularity frequency among all queries. The computational cost is $Com \cdot T$, where $Com$ is the cost of running the Oracle. Specifically, $Com = O(MN)$ if a dynamic programming algorithm is used.*

**Theorem 2.** *For Online Stream Cache Bandits algorithm ( Algorithm 2), without additional memory space, we set $\delta = 1/T$, then with $O(\log T)$ additional computational cost, the regret can be bounded by*

$$Reg(T) \leq (c_2 + \log T)T_0 + \frac{128\ell(M, \mathcal{Q})c_2}{p^*} \log^{\frac{3}{2}}(6TN)\sqrt{T \log T} \tag{7}$$

$$\leq O\left(\frac{\ell(M, \mathcal{Q})}{p^*} \sqrt{T} \log^2 NT\right) \tag{8}$$

*where $p^* = \min_{q \in \mathcal{Q}} P(q)$ is the minimum popularity frequency of all queries and $T_0 = \max\left(\frac{32}{p^*} \log TN + \frac{4}{p^*}, \left\lceil \frac{2ln(3TM)}{p^*} \right\rceil\right)$. On the other hand, for any caching policy $\{\mathcal{M}_t\}_{t=1}^{T}$, there exist some cases of $P(q), C^*(q)$ such that for some universal constant $C'$*

$$Reg(T) \geq C'\sqrt{T} \tag{9}$$

This proof is deferred to Appendix. Based on the theorems presented above, we demonstrate that Online Stream Cache Bandits achieves the same order of regret, $\tilde{O}(\sqrt{T})$, as Basic Batch Cache Bandits, but with a lower computational cost. In the work of Zhu et al. (2024), a lower bound of $\tilde{O}(\sqrt{T})$ is established. Our work extends their findings; consequently, the lower bound in our setting is naturally larger than theirs, making the proof of our lower bound evident. The regret established above conforms to $\tilde{O}(\sqrt{T})$, indicating that our work provides an optimal solution in relation to $T$.

Theorem 2 in Zhu et al. (2024) can be viewed as a special case in our setting, where each query has the same length. If we assume popularity follows a uniform distribution, the order of our regret over the time horizon $T$ will be $O(MN\sqrt{T})$, which aligns with the bound presented in Zhu et al. (2024).

---

**Algorithm 2** Online Stream Cache Bandits

---

**Require:**

$\mathcal{Q}_0 = \phi$. $T_0^{(p)}(q) = 0, q \in \mathcal{Q}$. $T_0^{(c)}(q) = 0, q \in \mathcal{Q}$. $\hat{P}_0(q) = 0, q \in \mathcal{Q}_t$ ($\sum_{q \in \mathcal{Q}} P(q) = 1$).

$\tilde{C}_0(q) = 0, q \in \mathcal{Q}$. $\hat{C}_0(q) = 0, q \in \mathcal{Q}$. $A(q) = 0, q \in \mathcal{Q}$. $cache_0 = \phi$. $\hat{cache}_0 = \phi$. Calling flag $i = 0$. The size of cache is $M$.

**Ensure:**

1: **for** round $t = 1, ..., T$ **do**
2:    A user arrive and select $q_t$ sampled from $\mathcal{Q}$ with the length of query is $L(q_t)$
3:    **if** $q_t \notin \mathcal{Q}_{t-1}$ **then**
4:        $\mathcal{Q}_t = \mathcal{Q}_{t-1} \cup \{q_t\}$
5:    **else**
6:        $\mathcal{Q}_t = \mathcal{Q}_{t-1}$
7:    **end if**
8:    $T_t^{(p)}(q_t) = T_{t-1}^{(p)}(q_t) + 1$
9:    $T_t^{(p)}(q) = T_{t-1}^{(p)}(q), \forall q \neq q_t$
10:   $\hat{P}_t(q) = T_t^{(p)}(q)/t, \forall q \in \mathcal{Q}_t$
11:   **if** $q_t \in \mathcal{M}_{t-1}$ **then**
12:       Output the result from cache
13:       $\hat{C}_t(q) = \hat{C}_{t-1}(q), \tilde{C}_t(q) = \tilde{C}_{t-1}(q), T_t^{(c)}(q) = T_{t-1}^{(c)}(q), \forall q \in \mathcal{Q}_t$
14:   **else**
15:       Agent input $q_t$ into LLM and receive the result of query with length $A(q_t)$, then observe the cost $c_1 \leq C_t(q_t) \leq c_2$.
16:       $\tilde{C}_t(q_t) = \tilde{C}_{t-1}(q_t) + c_t(q_t), T_t^{(c)}(q_t) = T_{t-1}^{(c)}(q_t) + 1$
17:       $\hat{C}_t(q) = \hat{C}_{t-1}(q), \tilde{C}_t(q) = \tilde{C}_{t-1}(q), T_t^{(c)}(q) = T_{t-1}^{(c)}(q), \forall q \neq q_t$
18:       $\hat{C}_t(q) = max\{c_1, \frac{\tilde{C}_t(q)}{T_t^{(c)}(q)} - (c_2 - c_1)\sqrt{\frac{2\log(6TN/\delta)}{T_t^{(c)}(q)}}\}, \forall q \in \mathcal{Q}_t$
19:   **end if**
20:   **if** $q_t \in \hat{\mathcal{M}}_{t-1}$ and $q_t \notin \mathcal{M}_{t-1}$ **then**
21:       Put $q_t$ and its process result which is from LLMs into cache.
22:       $\mathcal{M}_t = \{q_t\} \cup \mathcal{M}_{t-1}$
23:   **end if**
24:   **if** $t = 2^i$ **then**
25:       $\boldsymbol{w}_t = \{\hat{C}_t(q) \cdot \hat{P}_t(q)\}_{q \in \mathcal{Q}_t}, \boldsymbol{l}_t = \{L(q) + A(q)\}_{q \in \mathcal{Q}_t}$
26:       $Oracle(\mathcal{Q}_t, \boldsymbol{w}_t, \boldsymbol{l}_t, M) \to \hat{\mathcal{M}}_t$
27:       Clearing out the current cache $\mathcal{M}_t$
28:       $(\mathcal{M}_t = \mathcal{M}_{t-1} \cap \hat{\mathcal{M}}_t)$
29:       $i = i + 1$
30:   **else**
31:       $\hat{\mathcal{M}}_t = \hat{\mathcal{M}}_{t-1}$
32:   **end if**
33: **end for**

---

## 5 EXPERIMENT

### 5.1 SIMULATION DATASET

We conduct synthetic online experiments to evaluate our algorithm. In Fig.1, we compare performance of our algorithms and the baseline using cumulative regret in online learning with a simulated dataset. We use the method described in the appendix of Zhu et al. (2024) as a baseline, as it is the only algorithm we could find that is applicable to this issue. We consider 20 distinct queries and set the cache size to be 20. For the ground-truth, we model the popularity distribution as power distribution with $\alpha = 0.5$ and the expected of cost for each query is sample from a uniform distribution support on $[0.1, 1]$. We repeat the simulation 10 times and plot the mean and standard deviation in the figure. The cost for each query in each round is sampled from a truncated normal distribution

with a mean equal to the expected value generated above. We employ a dynamic programming algorithm as our Oracle.

As shown in Fig.1, both Basic Batch Cache Bandits and Online Stream Cache Bandits outperform baseline. This improvement can be attributed to the fact that the method in Zhu et al. (2024) provides an approximate solution to the knapsack problem (Zhou et al., 2008), which poses a risk of falling into sub-optimal solutions. In contrast, our algorithm circumvents this issue by utilizing the Oracle for the knapsack problem. We observe that Online Stream Cache Bandits performs worse than Basic Batch Cache Bandits. This discrepancy is due to the fact that Basic Batch Cache Bandits can be updated more promptly, allowing it to converge faster. Although Basic Batch Cache Bandits exhibits lower regret, as mentioned in Section 4, it requires substantial additional memory and computational resources, which may be unrealistic in real-world scenarios. Conversely, the Online Stream Cache Bandits algorithm achieves similar regret to Basic Batch Cache Bandits while avoiding these significant additional costs.

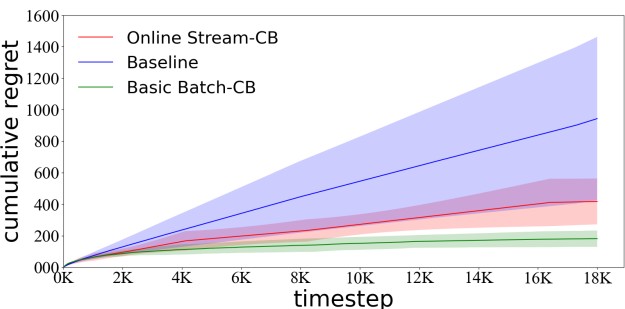

Figure 1: Synthetic dataset with 18000 rounds

## 5.2 REAL DATASET

In line with Zhu et al. (2024), we evaluate our algorithms using the OpenAssistant (Köpf et al., 2023) dataset for the chat assistant task. For this task, we employ the FastChat-T5-3B (Zheng et al., 2023) model to implement our online algorithm, utilizing inference latency as the cost metric. We run our algorithm with 100 distinct queries in the online setting over a total of 40,000 and 140,000 rounds, respectively, across three trials, with a cache length set to 100. Since our work focuses on cache selection, the quality of the responses is not a primary concern. As shown in Tab.1, our method reduced the cost by $12.4\%$ and $12.8\%$ at 40,000 and 140,000 rounds, respectively, compared to the baseline. After a sufficient number of online learning steps, the Oracle accurately learns the costs and frequencies of each query within this finite query pool, enabling Online Stream Cache Bandits to outperform the baseline algorithm.

Table 1: Cumulative cost on real dataset

| algorithm | cost(40000) | cost(140000) |
|---|---|---|
| Baseline | 6501.6 | 22602.6 |
| Online Stream Cache Bandits | 5692.5 | 19708.3 |

## 6 CONCLUSION

In this work, we explore a more generalized model applicable to the cache of Large Language Models (LLMs). We address the issue of variable-size caching and propose a streaming online algorithm to minimize additional overhead, demonstrating that our method performs better compared to previous work. Additionally, we provide a theoretical guarantee of $\tilde{O}(\sqrt{T})$ for our algorithm. By considering the knapsack problem to tackle the heterogeneous sizes of queries in real-world applications, we believe that this extension of cache will have a broader range of applications.

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
