# OpenReview forum: "Harnessing Query Heterogeneity for Cost-Effective Proactive Caching in LLM Inference"
_ICLR.cc/2025/Conference — Submitted to ICLR 2025_

### Official Review · Reviewer_AHQS · 2024-10-31

**Soundness:** 2
**Presentation:** 2
**Contribution:** 2
**Rating:** 3
**Confidence:** 4

**Summary:**

This paper studies the cost-effective caching problem in LLM inference. Increasing volume of query brings increasing resource consumption. Using caching to store important/frequent queries and answers can help to save the computational resources. This paper proposes a Knapsack bandit framework to characterize the learning process of unknown queries’ costs and find the optimal caching strategy. The paper proposes both a batched and online version of algorithm to solve the problem. The former with $O(T)$ computation cost and larger memory cost achieve $O(\sqrt{T}\log^{3/2}T)$ regret and the latter with $O(\log T)$ computation cost achieve $O(\sqrt{T}\log^{2}T)$ regret. Experiments on both synthetic and real-world datasets show advantage over baselines.

**Strengths:**

1.	The paper study a more general knapsack bandit problem for caching in LLM inference that considers the cost and weight differences over different queries.
2.	The paper proposes an Online Stream Cache Bandits algorithm to deal with the problem. The algorithm calls the oracle at most $O(\log T)$ times and can achieve sub-linear regret when learning the optimal caching strategy.

**Weaknesses:**

1.	In assumption 2, it has been assumed that each query would appear with probability at most ½. This assumption may be too strong. What would happen if this assumption is not satisfied?
2.	The regret depends on $1/p^*$ where $p^*$ is the minimum sampling probability among all queries. This term may be very large and dominate the result. Intuitively speaking, though there may exist a query with smaller sampling probability and it is hard to observe its cost, its influence to the final reward is also very limited. So in this case, the query does not need to be observed enough times as though queries with larger sampling probability, and thus the dependence on $1/p^*$ can be improved. Can authors comment on this?
3.	In Algorithm 2, the authors claim that it does not need to store all queries and answers. However, in Line 3-7, the algorithm still needs to check whether the query has been seen before. So it still needs to store all query information and pay additional memory cost.
4.	Line 205, typo.
5.	Line 409, ‘additional computational cost’ may bring confusion.
6.	In experiments Line 518, what does cache length mean? Does it refer to the number of queries that can store or the number of chars in the query?

**Questions:**

Please see the last part.

---

> ### Author Response · Authors · 2024-11-22
> **Response to reviewer AHQS**
>
> We thank all reviewers for their valuable comments.
>
> $\\textbf{Q1: Clarification on the Assumption 2.}$ Assumption 2 is primarily required for our theoretical analysis. The assumption that the sampling probability of each query is less than $1/2$ comes from Lemma 8 in [2], and it is used in our proof to theoretically bound the waiting time for Algorithm 2 to receive the required query. In practical applications, we treat certain hot topics as queries that are frequently repeated to the LLMs. The probability of these repeated queries typically does not exceed $1/2$. For example, in an e-commerce LLM, the proportion of the most frequently asked queries, such as refund requests, accounts for about 30% of all queries. While this represents a high probability in practice,  it is still less than $1/2$. On the other hand, it is clear that there can be at most one query with a probability greater than $1/2$, since we have $\\sum_{q\\in \\mathcal{Q}}P(q)=1$. We can easily identify this query and handle it separately, allowing the cache to still focus only on queries with a sampling probability less than $1/2$. In fact, based on our experiments, we found that whether this assumption holds does not significantly affect the performance of our experiments.
>
> $\\textbf{Q2: Regret depends on $1/p^*$.}$ In the online setting corresponding to Algorithm 2, when the Oracle provides the optimal cache solution, the required query may not be available immediately. As a result, the cache replacement cannot happen instantly; algorithm must wait for the required query to arrive before performing the replacement. Therefore, in the online setting, the term $1/p^*$ is difficult to eliminate due to the update delay. As we discussed in Section 4.3, in some special cases, such as the scenario in [1], $1/p^*$ can be replaced by $N$. On the other hand, if we assume popularity follows a uniform distribution, we have $p^*=1/N$ and the regret becomes $\\widetilde{O}(MN\\sqrt{T})$, which aligns with the bound presented in [1].
>
> $\\textbf{Q3: Memory cost in Algorithm 2.}$ We apologize for the misunderstanding in our expression. The precise expression should be that there is no need to store the answer information corresponding to all queries. Algorithm 1 needs to store the textual information of all queries and their corresponding answers, while Algorithm 2 does not require storing the textual information of all answers. The query’s textual information is typically used to allow the algorithm to recognize the sampled query and determine whether it is a cache hit. The textual information of the query’s answer enables the cache to directly return the saved answer text in the case of a cache hit, without requiring processing by the LLMs. Algorithm 1 requires storing the textual information of all answers to immediately replace query-answer pairs when updates occur. In contrast, Algorithm 2 only store the answer corresponding to the current query in the cache, without requiring storage for answers to queries not in the cache. At the same time, the textual information is much larger compared to the statistical data used for computation and updates, such as the count and estimate of each query. Therefore, reducing the memory usage of the textual information can help decrease the overall memory overhead. As a result, we can conclude that Algorithm 2 reduces memory costs. For the query textual information, we found that strategies using string hashing, such as BKDRHash [3], can be employed to create a hash table for the textual information of each query. By using this string hash function, the input string can be converted into a numerical value based on a specific encoding scheme, which can then be used to identify the current query, thereby reducing the storage required for the textual information.
>
> $\\textbf{Q4,Q5: typo.}$ We appreciate the reviewers for pointing out the typographical errors in our work. We will make the necessary corrections in the revised version.
>
> $\\textbf{Q6: Cache length.}$ This is a typographical error; it should be 'cache size', not referring to the number of queries that can store. In our paper, both "length" and "size" refer to the length of the query's textual information, which is also the memory size occupied by the query's text.
>
> [1] Banghua Zhu, Ying Sheng, Lianmin Zheng, Clark Barrett, Michael I. Jordan, and Jiantao Jiao. 2024. On optimal caching and model multiplexing for large model inference. In Proceedings of the 37th International Conference on Neural Information Processing Systems (NIPS '23). Curran Associates Inc., Red Hook, NY, USA, Article 2577, 59062–59094.
>
> [2] Li, S. and Zhang, S. (2018). Online Clustering of Contextual Cascading Bandits. Proceedings of the AAAI Conference on Artificial Intelligence, 32(1).
>
> [3] J. Wang, "Github," 9 8 2015. https://github.com/JamzyWang/HashCollector.

---

> > ### Comment · Reviewer_AHQS · 2024-11-23
> >
> > Thank you for the detailed response. I still have a concern regarding the dependence on \(1/p^*\). Such dependence is also observed in previous works on online influence maximization (OIM). In the OIM problem, certain edges can only be observed with a small probability \(p^*\), which causes existing algorithms to incur regret that scales with \(1/p^*\) (as discussed in Section 4.3 of [1]). However, subsequent works have demonstrated that, although these edges are observed with small probability and cannot be estimated accurately, their contribution to the regret is minimal due to the discounting effect of \(p^*\). Specifically, while these edges are only observed with probability \(p^*\) and estimated with an error of \(1/p^*\), their expected contribution to the regret remains comparable to other frequently observed edges when accounting for the occurrence probability \(p^*\) [2].
> >
> > In my view, the problem here is very similar to the one in OIM, and I think the dependence on \(1/p^*\) may not be necessary. Could you please clarify if there are fundamental differences in the settings that would justify this dependence?
> >
> >
> > [1] Combinatorial Multi-Armed Bandit and Its Extension to Probabilistically Triggered Arms. JMLR'16
> >
> > [2] Improving Regret Bounds for Combinatorial Semi-Bandits with Probabilistically Triggered Arms and Its Applications. NeurIPS'17

---

> ### Author Response · Authors · 2024-11-23
>
> Thanks for your valuable suggestions, this is a very meaningful question. Our problem shares similarities with OIM, particularly in treating the cache $M_t$ as a super arm. However, we identify key differences in action (super arm) selection between our online setting and OIM. In our online setting, the cache can only be updated using the currently arriving query and the queries already stored in the cache, imposing constraints on action selection. In contrast, OIM allows actions to utilize all base arms. Specifically, the key differences in action (super arm) selection are as follows:
>
> \begin{equation}
> M_{t+1} \\subseteq M_{t} \\cup \\{q_t\\} \\text{ (Ours)}
> \end{equation}
> \begin{equation}
> M_{t+1} \subseteq \mathcal{S} \text{ (OIM)}
> \end{equation}
>
> where $\\mathcal{S}$ is a fixed set of valid super arms for all queries.
>
> Namely, in our problem, action across different rounds exhibit local dependence, which makes the problem more challenging. As a result, the analytical methods and conclusions from [1,2] cannot be directly applied to our setting. Currently, it remains challenging for us to develop a clear method to eliminate the influence of $1/p^*$.
>
> [1] Combinatorial Multi-Armed Bandit and Its Extension to Probabilistically Triggered Arms. JMLR'16
>
> [2] Improving Regret Bounds for Combinatorial Semi-Bandits with Probabilistically Triggered Arms and Its Applications. NeurIPS'17

---

### Official Review · Reviewer_L88S · 2024-11-02

**Soundness:** 2
**Presentation:** 3
**Contribution:** 3
**Rating:** 6
**Confidence:** 3

**Summary:**

This paper considers the caching problem in LLM inference. Note that for a prompt, doing LLM inference to generate tokens can be costly. A way to reduce the cost is to do KV caching such that for the prompt stored in the cache, generating tokens will be costless. The paper formulates this procedure as a bandit problem. To be specific, the action is cache $M_t$. The paper assumes that the prompt arrives following an unknown distribution, generating a fixed inference cost when the prompt is not in the cache $M_t$ and generating zero cost when the prompt is in the cache $M_t$. Therefore, each possible cache is associated with an unknown expected cost. The paper takes a bandit learning approach to approximate the optimal caching. The whole procedure has $T$ rounds and at each round $t$, the decision maker estimates the costs, selects a cache $M_t$, and observe new information. Finally, the paper derives a $O(\sqrt{T})$ regrets bound of their algorithm and conducts numerical experiments to test the empirical performance of their algorithms.

**Strengths:**

1. The paper provides a bandit model to study the optimal caching problem and the model is more general than the models in the previous literature, such as Zhu et al. (2023). The model in the paper is more general and practical.

2. The paper provides a reasonable algorithm to approximate the optimal caching based on an online learning idea. The algorithm enjoys a strong $O(\sqrt{T})$ regret bound. The paper also conducts numerical experiments to illustrate the empirical performances of their algorithm.

**Weaknesses:**

1. The regret bound established in the paper depends on $1/p_{\min}$, where $p_{\min}$ is the minimum probability for a prompt to show up. However, this probability can be very small, especially when the set of possible prompts is very large. It is not known whether such a weak dependency on $p_{\min}$ is necessary or not.

2. The algorithms in the paper require an oracle to generate the optimal caching based on the current estimated costs. However, it is not discussed in the paper whether assuming the existence of such an oracle is practical or not. That is, what is the computational complexity of the oracle and what is the feasibility in real-world applications?

**Questions:**

1. Could you discuss whether the dependency on $p_{\min}$ is necessary or not? I think $1/p_{\min}$ can be very large.

2. Could you discuss what should be the oracle that outputs $M_t$? Is it a knapsack problem and is it easy to solve it?

---

> ### Author Response · Authors · 2024-11-22
> **Response to reviewer L88S**
>
> We thank all reviewers for their valuable comments.
>
> $\\textbf{Q1: Dependency on $1/p^*$ (or $1/p_{min}$).}$ Please refer to the detail discussion in our response to Q2 for Reviewer AHQS.
>
> $\\textbf{Q2: Oracle for $M_t$ and its relation to the knapsack problem.}$ We set the Oracle as a solver for the 0-1 knapsack problem. The Oracle is required to not only output the maximum value corresponding to the optimal knapsack but also provide the optimal knapsack itself. The algorithm then uses the optimal knapsack output by the Oracle to determine the current cache $M_t$ that should be selected. In our implementation, we adopt the classical dynamic programming approach and record the optimal knapsack found. The Oracle in our work provides the exact solution. While an approximate algorithm could be used to reduce computational overhead, it is crucial that the Oracle's solution remains stable. This is because, once the cache converges to an optimal state, any change in the Oracle's solution would trigger a cache update, forcing the system to converge to a new optimal cache. Such instability could prevent our algorithm from ever converging.

---

> > ### Comment · Reviewer_L88S · 2024-11-27
> >
> > I checked the author's response to Reviewer AHQS regarding the dependency on $1/p^*$ and it seems that a further convincing conclusion on whether $1/p^*$ is necessary or not is still lacking. Is there a formal lower bound showing that one cannot be better than $1/p^*$? I somewhat agree with Reviewer AHQS's intuition that although we may not estimate the part associated with $p^*$ accurately, since $p^*$ is so small, the influence is also small.

---

> ### Author Response · Authors · 2024-11-29
>
> Thank you for your feedback. Analyzing of the influence of $1/p^*$ remains a challenge. In the online setting of Algorithm 2, the cache can only be updated using the currently arriving query and the queries already stored in the cache, imposing constraint on action selection, i.e. $M_{t+1} \subseteq M_{t}\cup \\{q_t\\}$. Due to this constraint, the algorithm faces challenges in exploring other queries, and it becomes more difficult for the cache to recover from a suboptimal solution. Our lower bound is derived directly from previous work [1], which can be considered a special case of our model. If the query popularity distribution is uniform, then $p^*=1/N$, and our regret upper bound aligns with the result in [1]. Addressing the challenges associated with the term $1/p^*$ and providing a tighter lower bound is part of our future work.
>
> [1] Banghua Zhu, Ying Sheng, Lianmin Zheng, Clark Barrett, Michael I. Jordan, and Jiantao Jiao. 2024. On optimal caching and model multiplexing for large model inference. In Proceedings of the 37th International Conference on Neural Information Processing Systems (NIPS '23). Curran Associates Inc., Red Hook, NY, USA, Article 2577, 59062–59094.

---

> > ### Comment · Reviewer_L88S · 2024-11-29
> >
> > Thank you for the response! I will take it into consideration. Currently, I tend to maintain my original score.

---

### Official Review · Reviewer_BeGb · 2024-11-05

**Soundness:** 3
**Presentation:** 3
**Contribution:** 3
**Rating:** 6
**Confidence:** 2

**Summary:**

This interesting paper studies the caching of query responses for LLMs and addresses a few fundamentally different aspects of this problem namely: a) cost is not observed for already cached queries; b) cost varies by query (and they are of different sizes).  In order to solve this problem the authors note that: a) this is a bandit problem, but reward is only (re)observed at the time an item is placed in the cache b) the constrained optimization problem can be framed as a 0-1 knapsack problem.  Building upon the work of Zhu et al the authors propose two algorithms.  The first is a purely theoretical construct that applies a pessimistic LCB bandit formulation using an oracle to solve the knapsack problem, the second algorithm is an actually practically implementable algorithm which the authors show performs in more general settings to the algorithm of Zhu et al.  They provide regret bounds to support their simulation studies showing a O(root(T)) regret bound.

**Strengths:**

The content of this paper ventures outside areas of my expertise in a few respects, but I have a generally positive disposition to it.

The paper reads well, and to the best of my understanding addresses an important problem and makes an important contribution.  The main arguments and experimentation appear sound to me.  The scope is narrow and well-defined.

**Weaknesses:**

Some general comments / suggestions

I don't feel expert enough to comment in detail on the magnitude of the contribution.  While the problem formulation is interesting, I am doubtful it could be deployed as  real caching solution for LLMs due to some practical reasons (need to store cost and count of all queries, uncertainty on cost seems not too uncertain to me) - but am open to having the authors change my mind on these.

While the writing is technically quite good (with occasional lapses), I think the structuring of the overall argument could be improved.  Some interesting points are brushed over (e.g. LCB vs USB).  The core contribution is a bit of work to disentangle from the presentation.

This paper makes a contribution to cache bandits motivated by an LLM use case, I think the motivation could be altered to make the paper a bit more general.  That is explain cache bandits, what is known about them, what is lacking, and how current solutions are not applicable to LLM use cases.

Line 151, usually people consider contextual bandits to be a simplification of RL.
Line 215 opening quotes should be backticks
Line 374 and be more reasonable. -> and is more reasonable.
Line 455-458.  If I am not wrong it should be q_t not q in a few places.
Line 143.  Does the problem perfectly reduce to the knapsack problem or do simplifications need to be made?  Specifically why use the phrasing “we regard”, rather than “the problem reduces to”.

The differences between algorithm 1 and 2 are hard to see and understand, I would suggest colour might be useful to do a “diff” to help here.

**Questions:**

I would like to raise the following questions:
•	The use of pessimism as an exploration heuristic could be explained in more detail.  In a conventional setting UCB/optimism is based on: if an arm might be good select it, you will either receive a good reward, or learn quickly it isn’t good.  I can see that only observing a reward on the first arm pull might change this situation, but to me this argument isn’t fully developed.
•	I would expect the variation in the computational cost to be quite small in practice, i.e. the variance in epsilon, so I wonder about the utility of averaging over multiple observations of the reward.
•	Algorithm 2 is presented as being a more practical version of Algorithm 1, but it still requires maintaining a count of how many times each query was observed and it’s cost.  Is this really realistic?
•	Does your work provide any guidance on how large the cache should be?

---

> ### Author Response · Authors · 2024-11-22
> **Response to reviewer BeGb**
>
> We thank all reviewers for their valuable comments.
>
> $\\textbf{Clarification on problem reduction to knapsack problem.}$ We appreciate the valuable feedback from the reviewers. Our problem setting can indeed be reduced to the 0-1 knapsack problem. We will correct this statement in the revised version.
>
> $\\textbf{Q1: More detailed explanation of pessimism method.}$ As mentioned in Section 1, we use the lower confidence bound (LCB) method for estimation to prevent our cache policy from getting trapped in a suboptimal solution, a method also used in previous work [1]. The key idea behind LCB is to penalize queries that remain in the cache for too long, making them more likely to be replaced. For example, consider a query set $\\mathcal{Q}=\\{1, 2\\}$ with a cache size of one. If the optimal cache is $\\{1\\}$, and the algorithm inserts $2$ into the cache, the algorithm will always replace the query with the higher estimated value. In this case, only queries not in the cache have their cost estimates updated due to a miss, while those in the cache retain their estimates. Under LCB, when $1$ experiences a miss, its estimation increases as the confidence interval narrows, eventually surpassing $2$ and leading to its replacement. In contrast, with traditional UCB (optimism method), $1$'s estimate would decrease with each miss, always remaining lower than $2$, and thus never being replaced. This situation is referred to as the algorithm getting trapped in a suboptimal solution. A detailed discussion can be found in Section 4.1.
>
> $\\textbf{Q2: Utility of averaging over multiple reward observations.}$ In our work, we primarily considered the impact of cost rather than reward, aiming to enhance the overall reward by reducing cost, so we did not consider the observation of reward here. Since cost and reward are complementary, we believe that optimizing for reward can also be applied to the problem we propose.
>
> $\\textbf{Q3: The practicality of maintaining query count and cost.}$ Compared to Algorithm 1, Algorithm 2 primarily reduces memory usage associated with the textual information of the query's answer. In practice, the memory required to store the textual information of each query's answer is much larger than the memory needed to record the observation count of each query and store statistical information, such as its estimation and count information. Therefore, we assert that Algorithm 2 reduces memory overhead compared to Algorithm 1, making it more practical for real-world applications. On the other hand, storing the count and statistical information of queries is essential in bandit strategies, as this information is needed for the policy to compute and update effectively. Moreover, the memory overhead for this part is small.
>
> $\\textbf{Q4: Guidance on cache size.}$ The size of the cache can be decided based on the user's actual needs, but it should be much smaller than the total size of all queries, as in practical applications, the query set can be very large. A larger cache can reduce the overall cost, but a larger cache also incurs greater additional storage overhead. On the other hand, if the cache is too small, it may result in insufficient improvement for reward. In our simulation experiments, we set the cache size to be twice the maximum value of the sum of the query length and its corresponding answer length, i.e. $2\\max_q{S(q)}$.
>
> [1] Banghua Zhu, Ying Sheng, Lianmin Zheng, Clark Barrett, Michael I. Jordan, and Jiantao Jiao. 2024. On optimal caching and model multiplexing for large model inference. In Proceedings of the 37th International Conference on Neural Information Processing Systems (NIPS '23). Curran Associates Inc., Red Hook, NY, USA, Article 2577, 59062–59094.

---

> > ### Comment · Reviewer_BeGb · 2024-12-01
> > **Thanks for the clarifications**
> >
> > Q1.
> > Thanks for the further explanation.  At least for me it remains hard to understand, I appreciate the effort to address this subtle issue in Section 4.1, but the issue for me still remains opaque.  Given that it's the most interesting part of your paper I would expand your example and add it to the paper.  The fact that UCB is seen as a method to trade-off explore exploit, so switching to LCB is unexpected is not addressed clearly IMO.
> >
> > Q2
> > I should have said multiple cost observations or value observations not multiple reward observations.  I would expect the cost to have low variance (maybe we could safely assume there is no variance).
> >
> > Q3
> > I repeat my suggestion that showing a diff of the two algorithms side by side could be enlightening.  I understand storing query counts are essential in bandit strategies, but if the number of queries is astronomical in size (almost every query is unique) then it might still be infeasible.
> >
> > Q4.
> > Thanks for your interesting response.

---

> > > ### Author Response · Authors · 2024-12-02
> > >
> > > Thank you for your valuable comments.
> > >
> > > Q1. In our setting, an arm is pulled only when there is a cache miss. This means we are effectively optimizing the portion of the system outside the cache, aiming to minimize the cost associated with cache misses. When the objective is to minimize the cost of cache misses, using the LCB strategy is more appropriate. To put it simply, when the goal is to maximize the optimization objective, the UCB strategy is typically used. In contrast, when the goal is to minimize the objective, the LCB strategy is more suitable.
> > >
> > > Q2. In our setting, we assume that the noise of cost is a subgaussian distribution. The assumption of subgaussian noise is a standard assumption in the bandit literature. On the other hand, [1] discuss several possible choices of cost: FLOPS, latency of the model, the price for API calls, user satisfaction of the results, or a combination of all the four factors. The cost in our setting can be influenced by various factors, making it a random variable. If the cost were fixed, simpler strategies could be employed, such as submitting all queries to the LLM at once to obtain a deterministic cost without the need for estimation. This scenario, where the cost is constant, represents a more basic setup and can be considered a special case of our problem. However, such a setup is less versatile and lacks the applicability of our approach, which accounts for randomness and variability in the cost, allowing for greater flexibility and more robust handling of real-world scenarios.
> > >
> > > Q3. Thank you for your valuable suggestions. The case of having a large number of arms is a challenging issue in the bandit field. However, this is not the focus of our work, so we have not considered the scenario of a large-scale number of arms in this study. On the other hand, the number of frequently asked queries over a period of time is relatively small compared to all queries. We can design our strategy to focus only on this subset of hotspot queries, so the number of arms in our problem will not be excessively large.
> > >
> > > [1] Banghua Zhu, Ying Sheng, Lianmin Zheng, Clark Barrett, Michael I. Jordan, and Jiantao Jiao. 2024. On optimal caching and model multiplexing for large model inference. In Proceedings of the 37th International Conference on Neural Information Processing Systems (NIPS '23). Curran Associates Inc., Red Hook, NY, USA, Article 2577, 59062–59094.

---

> > > > ### Comment · Reviewer_BeGb · 2024-12-02
> > > > **quick response**
> > > >
> > > > Q1 I don't think it's as simple as this, as exploration in this setting requires an item to move in and out of the cache repeatedly.  Where in a standard bandit exploration occurs every time an arm is pulled.
> > > >
> > > > Q2 I don't doubt that there is some randomness, but I find it hard to understand why the variance would be anything but tiny.

---

> > > > > ### Author Response · Authors · 2024-12-02
> > > > >
> > > > > Thank you for your valuable comments.
> > > > >
> > > > > Q1. Our problem differs from traditional bandit problems, as mentioned in Section 3 of [1]. In our setting, feedback is only generated for queries that are not already in the cache, meaning that only unselected arms receive feedback. This contrasts with the standard bandit setup, where all arms (queries) can receive feedback. In our case, queries that are already in the cache do not generate feedback due to cache hits. Under the LCB strategy, the estimated values for these queries with missing feedback are biased downward, and the estimated values for cache-missing queries will increase as feedback accumulates. This process allows queries in the cache to be replaced, effectively updating the cache, as demonstrated in the example provided in our first response.
> > > > >
> > > > > Q2. In our setting, we assume that the noise of cost $\epsilon$ follows an $R$-subgaussian distribution. From this, we can derive that the variance of $\epsilon$, denoted $Var[\epsilon]$, is bounded by the constant $R^2$. This assumption is standard in the bandit literature. Therefore, a cost with very small variance can be considered a special case of our problem. Additionally, it is typically assumed that $R=1$ (which we also adopt) to simplify theoretical analysis. If $R\ne 1$, it does not affect the performance of our algorithm, and we would simply need to multiply the regret bound by an additional factor of $R$. In fact, in many fields such as communication, control, and others, noise is often assumed to follow a Gaussian distribution, especially in the absence of detailed noise models. The Gaussian distribution is a special case of the sub-Gaussian distribution. This assumption helps us to effectively simplify the problem. In our experiments, we choose model latency—the time taken by the model to answer a query and return the response to the user—as the cost. In the actual use of LLM models, we have observed that the variation in this latency is not significant.
> > > > >
> > > > > [1] Banghua Zhu, Ying Sheng, Lianmin Zheng, Clark Barrett, Michael I. Jordan, and Jiantao Jiao. 2024. On optimal caching and model multiplexing for large model inference. In Proceedings of the 37th International Conference on Neural Information Processing Systems (NIPS '23). Curran Associates Inc., Red Hook, NY, USA, Article 2577, 59062–59094.

---

### Official Review · Reviewer_R5Gr · 2024-11-10

**Soundness:** 2
**Presentation:** 3
**Contribution:** 3
**Rating:** 6
**Confidence:** 4

**Summary:**

The paper studies the problem of efficient LLM inference through caching the seen queries with online learning. It focuses on the setting when the query legnth can be different.

**Strengths:**

The paper is built up on Zhu et al 2024, but proposes some new modifications that deal with
(1) variable query size, which is more realistic in real-world settings;
(2) the case when the length of the query is on par with the length of the cache size.

The authors provide new algorithms that guarantee near-optimal rate for the case of variable query length.

**Weaknesses:**

It's a little unclear to me how the main contirbution of the paper differs from that of Zhu et al. 2024. In particular, it seems that the lower bound directly applies here since Zhu et al. 2024 can be viewed as a special case of the condition the authors considered?

**Questions:**

1. For the presentation of lower bound, why are the authors using $O$ but not $\Omega$?
2. The length of the output for an LLM is usually non-determinisitic since LLM often samples with a non-zero temperature. So it seems that $A(q)$ here shall be a random variable. How does the algorithm account for it?
3. Given that the tightness w.r.t. $T$ is clear from prior work, can the authors also provide comments on the tightness with respect to the other parameters?

---

> ### Author Response · Authors · 2024-11-22
> **Response to reviewer R5Gr**
>
> We thank all reviewers for their valuable comments.
>
> $\\textbf{Clarification on contributions.}$ Our main contribution is that we extend [1] to capture query size, which enables us to design an algorithm that addresses this problem and analyze its regret with variable query sizes. The involving of variable query size presents several challenges in our theoretical analysis, such as we can not determine how many queries are currently stored in the cache or to find the regular of value of the queries stored in the current cache. Despite these challenges, we demonstrate that our Online Stream Cache Bandits algorithm achieves a $\\tilde{O}(\\sqrt{T})$ regret bound with a $O(\\log T)$ additional computational cost. Since our work is an extension of [1] and the problem setting in [1] can be considered as a special case of ours, we directly used the conclusion of [1] as our lower bound.
>
> $\\textbf{Q1: Notation mistake.}$ We apologize for the notation error in our work; it was a notation mistake, and the correct notation should be $\\Omega$. We will make the necessary correction in the revised version.
>
> $\\textbf{Q2: Handling Randomness in $A(q)$ due to Non-Deterministic LLM Outputs.}$ 1) In our setting, we assume that the length of each response to a query is fixed, as we focus on tasks with stable answer lengths for LLMs, such as machine translation and text classification. Generally, the answers from LLMs do not change significantly within a over time, so we primarily consider the case where the length of the answer remains fixed. 2) In cases where the size of answer to a query is random, [1] provides a brief discussion but no detailed analysis. We believe that the expected value of the answer size can be estimated and that this expected length can be used to define the optimal cache. To address random size, a good method is to use the expected value of the query length as the input to the Oracle and update the cache accordingly.
>
> $\\textbf{Q3: Tightness with respect to parameters other than $T$.}$ The analysis for the tightness of other parameters in Algorithm 2 remains a challenge. In the online setting, the cache can replace at most one query per round, leading to delays when updating the cache to the recommended version. These delays are difficult to bound, so we have only conducted a rough analysis of the lower bound. As a result, we are currently unable to provide precise lower bound, but we plan to address this issue in future work.
>
> [1] Banghua Zhu, Ying Sheng, Lianmin Zheng, Clark Barrett, Michael I. Jordan, and Jiantao Jiao. 2024. On optimal caching and model multiplexing for large model inference. In Proceedings of the 37th International Conference on Neural Information Processing Systems (NIPS '23). Curran Associates Inc., Red Hook, NY, USA, Article 2577, 59062–59094.

---

### Meta-Review · Area_Chair_EbF5 · 2024-12-17

**Metareview:**

This paper studies caching of LLM queries of variable sizes and unknown costs. The key idea is to greedily cache queries with highest lower confidence bounds (LCBs) on their costs. If a query is suboptimal and cached, it gets replaced by a more costly query whose estimated cost improves over time due to not being cached. The authors study two variants of the problem, propose algorithms for them, analyze them, and evaluate them empirically. The scores of this paper are 3x 6 and 3, which is an improvement over the initial 2x 6, 5, and 3. The reviewers had several concerns:

* **Algorithm design:** The algorithm design needs to be better explained. For instance, the authors use LCBs while the standard approach in exploration are upper confidence bounds (UCBs). I also wonder why the query probabilities are estimated using their empirical means and not LCBs. Finally, the authors need to elaborate on details of how the pessimism helps. The reason is that query caching is a complex optimization problem. Therefore, it is not immediately clear when better estimated optimal queries that are not cached replace cached suboptimal queries.

* **Theory:** The regret bounds in Theorems 1 and 2 are $O(1 / p_*)$, where $p_*$ is the minimum query probability. Therefore, they are loose. A matching lower bound clearly does not exist. Why? When the probability of the least probable query goes to zero, the query is unlikely to appear and thus the problem becomes easier.

* **Technical novelty:** The technical novelty in the proofs is unclear.

These concerns require a major revision and therefore the paper cannot be accepted at this time.

**Additional Comments On Reviewer Discussion:**

See the meta-review for details.

---

### Decision · Program_Chairs · 2025-01-22

Reject